# Impact of a Three-Year Obesity Prevention Study on Healthy Behaviors and BMI among Lebanese Schoolchildren: Findings from Ajyal Salima Program

**DOI:** 10.3390/nu12092687

**Published:** 2020-09-03

**Authors:** Carla Habib-Mourad, Lilian A. Ghandour, Carla Maliha, Michèle Dagher, Samer Kharroubi, Nahla Hwalla

**Affiliations:** 1Department of Nutrition and Food Sciences, Faculty of Agriculture and Food Sciences, American University of Beirut, Beirut 11-0236, Lebanon; ch18@aub.edu.lb (C.H.-M.); cm35@aub.edu.lb (C.M.); mad24@mail.aub.edu (M.D.); sk157@aub.edu.lb (S.K.); 2Department of Epidemiology and Population Health, Faculty of Health Sciences, American University of Beirut, Beirut 11-0236, Lebanon; lg01@aub.edu.lb

**Keywords:** childhood obesity, nutrition policies, Body Mass Index, schoolteachers, school health promotion

## Abstract

Most school-based obesity prevention programs in low- to middle-income countries are of short duration, and few undertake follow-up analyses after the termination of the project. The aims of the current study are to investigate (1) the long-term effects of a school-based intervention program when implemented over two years on body mass index (BMI), healthy dietary behaviors, and physical activity (PA); and (2) whether the effects are sustained after one-year washout. The study is a cluster-randomized trial; 36 public and private schools were randomized into either intervention or control groups. Students (8–12 years) completed pre-and post-assessment anthropometric measurements and questionnaires about their eating and physical activity habits. Students in the intervention groups received the program components for two consecutive years. Multiple logistic regression models were used to examine the effect of the intervention on BMI and healthy behaviors. Students in the intervention groups were less likely to be overweight at washout, only in public schools. The number of children reporting change in dietary behaviors significantly increased in intervention groups, with a sustained effect only in public schools. Policies aiming at securing a positive nutrition environment in schools, and adoption of nutrition programs, are needed for achieving sustained behavior and prompting BMI changes in children.

## 1. Introduction

Overweight and obesity in children and adolescents is an escalating global public health problem. More than 60 million children around the world are identified as obese with the highest prevalence in North America and the Middle East [1]. Obesity rates increased from less than 1% in 1975 to nearly 6% in girls and 8% in boys in 2016. Combined, the number of obese five- to 19-year-olds rose more than 10-fold globally, from 11 million in 1975 to 124 million in 2016 [2]. Childhood obesity rates continue to rise in some countries, such as Mexico, India, China and Canada, while it is slowly plateauing in other countries including United States (U.S.) and United Kingdom (UK) [2]. In Lebanon, a small country in the Middle East, trends of obesity from two national cross-sectional surveys conducted in 1997 and 2009, showed a statistical increase in obesity risk among children and adolescents in 2009 as compared to 1997, particularly among 6- to 19-year-olds. The odds of obesity was approximately 2 times higher in 2009 as compared to 1997 (45.2% and 20.0%, respectively) [3]. The increase in obesity prevalence in children had been mostly attributed to the rise in the prevalence of associated behavioral risk factors such as unhealthy eating habits and low physical activity particularly in countries undergoing the nutrition transition [4]. This is the case of Lebanon, where access to affordable energy-dense and westernized food products has increased in the last few decades [5]. Westernized dietary patterns have been linked nationally with an increased risk of overweight among adolescents [6]. In addition to dietary risk factors, researchers have found that, among Lebanese adolescents (14–18 years old), total screen time including television viewing, computer and videogames, ranged between 26.5 and 33 h/week thus exceeding values reported for US adolescents, which ranged between 15.3 and 21.2 h/week [7].

Internationally, preventing childhood obesity and addressing dietary and sedentary behaviors has been achieved through multi-component school-based interventions. Reviews on the effect of different types of school interventions showed that multi-component interventions seem to have superiority over single component interventions in adiposity reduction and yields many health benefits other than adiposity reduction. Successful outcomes on dietary behaviors, activity level and/or weight changes were observed particularly when interventions were developed within health behavior theories and strategies such as family involvement, interactive learning, and policies to alter the school food environment [8,9,10]. In fact, evidence was strongest for school-based studies that included a home and community component and implemented a combined diet and physical activity intervention [11]. Additional reviews showed that interventions that focus on reducing screen time or increasing time spent performing moderate to vigorous physical activity may also be particularly beneficial [12]. The effectiveness of school-based obesity interventions have been investigated and confirmed in populations within North America, Australia and Europe [10]. In the Middle East, however, where obesity rates are on the rise [13], only one study from our research team has investigated the impact of a multicomponent intervention (Health-E-PALS), which aimed at increasing nutritional awareness, improving eating habits and active lifestyle among 9–11 year-old schoolchildren to curb escalating adolescent obesity [10]. Health-E-PALS significantly improved children’s dietary behaviors and increased their nutritional knowledge and self-efficacy; however, obesity indices did not change after one year [14]. Recent research showed convincing evidence that long-term (more than one year) school-based obesity prevention interventions may be effective in reducing body mass index (BMI) in children [15]. In the absence of studies investigating the impact of longer-term interventions on child and adolescent obesity in the Middle East region, we conducted another study to investigate the two-year impact of the Health-E-PALS intervention, as well as its sustainable impact after a one-year washout, the name of this intervention program was modified to Ajyal Salima which means “Healthy Kids” in English. The aims of the study were specifically: (1) examine the impact of the two-year intervention on children’s BMI, dietary habits and physical activity; and (2) investigate the sustained impact on the same outcomes one year following the completion of the intervention. The paper provides evidence-based strategies for sustainable school-based interventions that can be adopted or adapted by countries of the Middle East region to curb the escalating burden of obesity.

## 2. Materials and Methods

### 2.1. Participants, Recruitment and Procedures

The current study is a three-year stratified cluster randomized intervention, including a two-year intervention program followed by a one-year washout period. The study was conducted in Beirut, capital of Lebanon, during the years 2014–2017.

Public and private primary schools were conveniently sampled. Both private and public schools were chosen to ensure the inclusion of a diverse group of students with various socioeconomic statuses (SES), since middle- to high-income families in Lebanon tend to enroll their children in private schools with high annual tuition fees, whereas lower income families tend to send their children to public schools for a nominal fee. Private schools were directly approached by the research team to participate in the study whereas public schools were recruited by the Lebanese Ministry of Education and Higher Education (MEHE). The final list of participating schools included 20 public and 16 private schools. Schools were stratified by type (private and public); within each stratum, schools were randomized into either receiving the intervention or acting as a control school. Within each participating school, all classrooms in grades 4 and 5 (aged 8–12 years) were approached, and all students in the selected classrooms were invited to participate in the study. Consent forms were sent to the students’ parents/guardians to obtain their approval; students also signed assent forms. Forms were sent to 2000 students and their parents, of which 1239 agreed to participate. Only students whose parents had consented and who had themselves assented participated in the study. A total of 698 students were ultimately assigned to receive the intervention and 541 to serve as the control group. Students in intervention schools received the program components for two consecutive academic years (within eight months each year). Twelve nutrition education interactive activities were delivered in the classroom during the first academic year and six complementary activities were delivered during the second academic year. The third year was the washout year (no intervention was administered). Students in the control schools did not receive any intervention through the entire three-year study period. After completion of the study, students in the control schools were offered the opportunity to receive the intervention. Figure 1 summarizes student recruitment, sample retention and data collection in both groups from baseline to study completion.

The students who were lost to follow-up at the end of the third year (washout year) were no different from those who stayed in the program on all covariates (data available upon request).

This study was retrospectively registered at Clinicaltrials.gov, on 5 March 2020 (NCT04297059). URL: https://clinicaltrials.gov/ct2/show/NCT04297059?term=ajyal+salima&draw=2&rank=1.

### 2.2. Ethics Approval and Consent to Participate

The intervention was granted ethical approval by the Institutional Review Board (IRB) of the American University of Beirut (project identification code NUT.NH.16).

Consent forms were sent to the students’ parents/guardians to obtain their approval; students also signed assent forms. Ultimately, only students whose parents had consented and had themselves assented participated in the study.

### 2.3. Data Collection

Baseline assessment was conducted for all students (in intervention and control schools) during October 2014. Thereafter, students in the intervention group received the intervention during the two academic years 2014–2015 and 2015–2016 and the academic year 2016–2017 was the wash-out year. Two post-assessments were conducted. The first post-assessment at the end of the second year (May 2016), and the second after the one-year washout period (May 2017).

Anthropometric measurements including height and weight were carried out by trained fieldworkers using standardized techniques and calibrated equipment. The measurements were conducted for each student separately and privately in the school nurse’s office. Subjects were weighed to the nearest 0.1 kg using a calibrated balance (Seca model 11,770, Seca GmbH & Co., Hamburg, Germany) in light clothing and with bare feet or socks/stockings. Height was measured with a stadiometer without shoes and recorded to the nearest 0.5 cm. BMI was calculated as weight divided by height (kg/m^2^). As per the World Health Organization (WHO), BMIs were converted to z scores based on gender and age, and then categorized as per WHO cut-off points [16]. Age was calculated based on birth date and date of examination.

The dietary and physical activity questionnaire was adapted from a previously developed tool used in Lebanese children [17]. The content validity of the original survey instrument was confirmed by a panel of experts consisting of one epidemiologist, one pediatrician and one nutritionist. The questionnaire was first developed in English, then translated to Arabic, and back-translated to English. The original and back translated versions were reviewed for consistency in meaning by two bilingual experts. Prior to the intervention, the questionnaire was pilot tested with 25 children aged 9–11 years (not included in the study), to ensure that all questions were relevant and well-understood. Relevant modifications were made after the testing. The dietary habits questions assessed intake of fruits, vegetables, and snacks (e.g., Do you usually eat fruits?); responses included yes, twice or more per day; yes, once a day; yes, but not every day; and no). Physical activity questions assessed regular sessions of physical education at school, and after school physical activities (e.g., Do you play or practice activities after school or during weekends?); responses were: yes, 1 time per week; yes, 2 times per week; yes, 3 times per week; yes, everyday. Categorical variables were recoded to binary formats in a manner that reflected nutritional guidelines for children [18]. The nutrition knowledge questions were coded as 1 for a positive answer or 0 for a negative answer (including ‘don’t know’ responses). The summed score (range: 0–14) reflected overall level of knowledge (the higher the score the better the knowledge). The self-efficacy questions tested students’ self-confidence in their ability to choose healthy foods and increase their physical activity (e.g., How sure are you that you can prepare healthy snack/breakfast eat more fruits/vegetables per day; or do more sports during the week?). The 9 self-efficacy items were also summed into a single score (range 0–18); each question was measured on a 3-point Likert scale (0 = not sure, 1 = little sure, 2 = very sure); the higher the score, the higher the self-efficacy. The internal consistency (and item-total correlations) of each set of knowledge and self-efficacy items was checked prior to summation (Cronbach’s alpha value = 0.66 at pre-assessment and 0.7 at post-assessment). The same questionnaire was used at all-time points. Trained fieldworkers administered the questionnaires to students in their usual classrooms. Students self-completed the questionnaires in the presence of the research team and absence of any school personnel.

### 2.4. Process Evaluation

Process evaluation was carried out all throughout the study to explore the fidelity of intervention implementation by teachers, examine their view and investigate contextual factors that affected the delivery of the intervention in public and private schools. Data collection was through direct researchers’ observation and field notes during each school visit. All schools were assessed during weekly fidelity checks. The checklist included questions regarding the intervention dose, integration of the activities in different class courses, the use of the intervention educational material and the cooperation of all teachers in the program implementation. It also assessed the improvement of the available food items in the school shops. The findings of the process evaluation are qualitatively summarized in this paper.

### 2.5. Intervention Design

The intervention focused on the promotion of healthy eating and active lifestyle. Its specific objectives included increasing fruit and vegetable consumption to at least five a day as well as breakfast and healthy snacks intake; controlling high-energy dense foods and beverages consumption; encouraging regular physical activity; and reducing time spent in sedentary activities to less than two-hours a day. The intervention was based on the constructs of social cognitive theory [19], and comprised three coordinated components. The first component consisted of culturally appropriate classroom sessions using fun and interactive activities delivered once a week by teachers who had received a ‘training of trainers (ToT)’ workshop on all program components, as well as hands-on coaching on all educational activities by the research team.

The intervention sessions provided appropriate nutrition education in a simple and fun layout. Each session consisted of two sections; 10 to 15 min of discussion, information and interaction about the topic of the week followed by 30 min of activity: game and/or food preparation. A set of attractive visual aids were distributed to students; the kit consisted of posters, pamphlets, activity booklets, cards and board games.

The second component involved parents included meetings, health fairs, where the program was introduced to families and to assist them in creating a supportive environment at home for healthy lifestyle behaviors. Healthy meals were offered following the meetings. Take-home packets summarizing the major points covered during the educational sessions were also sent home along with some food samples and recipes. The goal of the take home pamphlets was to address non-compliance and poor attendance of parents’ school meetings.

The third component included a food service intervention targeting the school shops and the lunch boxes sent by the family.

When the program was scaled up and rolled out as Ajyal Salima–Healthy Kids, the Ministry of Education and Higher Education in Lebanon adopted it to become a mandatory component in the public schools’ health curriculum which facilitated its implementation.

### 2.6. Data Analysis

Exploratory data analyses was conducted using Stata version 14. Frequency measures (counts and percentages) were computed for all categorical outcomes, and mean and standard errors for continuous outcomes. Independent sample *t*-tests and Pearson’s chi square were used to compare intervention and control groups on several continuous and categorical variables, respectively. For chi-square analyses, ‘clchi2′ was used as this program calculates cluster-weighted chi-square values for comparing dichotomous estimates when the unit of randomization is a cluster (in this case, cluster = school) [20].

The impact of the intervention on body weight was examined in two ways. In the first method, the changes in BMI z-scores after the completion of the intervention and one year after washout were computed using the following formula: Change in BMI z-score = current BMI z-score − BMI z-score at baseline.

Generalized estimating equations (GEEs) analysis was also used to compare the mean changes in BMI z-scores between intervention and control, controlling for age and BMI z-scores at baseline. The second method estimated the odds of being overweight/obese versus not, using multiple logistic regression, adjusting for age and baseline BMI.

Multiple logistic regression models was also used to estimate the effect of the intervention on binary measures of dietary behaviors in addition to other outcomes (e.g., physical activity, self-efficacy), whilst controlling for age, gender and baseline measures.

All analyses were carried out for the whole sample, in private and public schools. Analyses accounted for clustering at the level of the school (cluster = school). The critical alpha was set at 0.05.

## 3. Results

### 3.1. Sample Characteristics

Table 1 describes the general characteristics of the study population. The mean age of participants was 9.95 ± 1.13 years (8–12 years), with children in the intervention group being significantly younger than those in the control group (*p* < 0.01). As for gender, the study sample was distributed as 46.3% males and 53.7% females with no significant difference between intervention and control groups. Similarly, there was no significant difference in the distribution of the sample across the private and public schools between intervention and control groups.

As for BMI, while 59% of the total sample had a normal BMI at baseline, 22.7% were at risk of being overweight, 14.6% were overweight and 3.2% were obese. These proportions were comparable between the intervention and control groups; students belonging to both groups were also very similar in their baseline behaviors and habits, except for chips consumption (Table 1).

### 3.2. Impact of Intervention on Body Mass Index (BMI) and Obesity

Looking at the mean BMI z-scores (Table 2), there was no statistical difference between intervention and control groups post intervention and after washout (post intervention: 0.07 ± 0.05 vs. 0.145 ± 0.05, *p* = 0.27, and after one-year washout: 0.134 ± 0.05 vs. 0.237 ± 0.05, *p* = 0.16), for the overall sample. Similarly, no changes were observed when considering public and private schools separately (Table 2).

Table 3 shows the intervention effect on adiposity values post intervention and after one year-washout. The odds of being overweight/obese post intervention was similar in intervention and control groups, controlling for age and gender, and baseline BMI status. After one-year washout, changes were observed only in public schools, whereby students in the intervention group had a 52% reduced odds of being overweight/obese compared to students in control group.

### 3.3. Impact of Intervention on Dietary Habits

Table 4 shows the results of the multiple logistic regression analyses exploring the effect of the intervention on changes in dietary habits post intervention and after one-year washout, controlling for baseline measures of dietary behaviors and age. In the private schools, statistically significant increases in the odds of consuming fruits (odds ratio (OR): 1.84, 95% confidence interval (CI): 1.30; 2.60) and raw veggies (OR: 2.32 95% CI: 1.42; 3.80) were observed post intervention. Similarly, in public schools, the odds of fruits and raw veggies consumption was significantly higher in intervention versus controls, after washout; 1.66 and 2.19 times higher respectively. As for chips consumption, a statistically significant reduction was observed in public schools post intervention, with odds being 56% less in intervention versus control schools. After washout, the odds of chips consumption was still 30% less in intervention versus control public schools but not statistically significant (*p* = 0.06). In private schools, the odds of consuming chips post intervention and after one-year washout, did not differ between intervention and control groups. Finally, the odds of having soft and sweet drinks significantly decreased respectively (*p* = 0.027; *p* = 0.032) in intervention compared to control groups in the total sample.

### 3.4. Impact of Intervention on Physical Activity

Levels of PA after school did not change after two years of intervention. However, after one-year washout, the increase in PA was significantly higher in intervention schools versus control in the total sample (Table 4).

### 3.5. Impact of Intervention on Knowledge and Self-Efficacy

Table 5 presents the mean differences in knowledge and self-efficacy scores between baseline and two years post intervention, and between baseline and one-year washout, for intervention and control groups, separately for private versus public schools, after adjustment for baseline measures, age and gender.

In both public and private schools, and after two years of intervention, knowledge scores increased and were significantly higher in the intervention group versus control (*p*-value < 0.0001) with a sustained effect after one-year washout (Table 5).

With regard to self-efficacy scores, the latter significantly increased post-intervention in intervention versus control schools in the total sample, as well as in public schools and private schools. After one-year washout, scores slightly increased in the intervention groups and worsened for the control hence a statistically significant difference between the two groups was observed in the total sample and in public schools, but not in private schools (Table 5).

### 3.6. Process Evaluation Outcomes

Data synthesized via informal conversations and interviews with teachers, students and school principals showed the following:Classroom activities: during the first year, the program was delivered as designed in both public and private schools. The 12 educational lessons and activities were all implemented in the intervention schools along with meetings with parents and the food service management. In the second year, the commitment of teachers in private schools decreased, thus reducing both the reach and dose received by the students; while it remained constant in public schools given the adoption of the program by the Ministry of Education.School shop: amendments at the school shops level were more successful in public schools, since the MEHE has issued a ministerial order regulating the type of snacks and drinks to be sold. The sale of energy dense snacks (e.g., chips, donuts, croissant, hot dogs, sweet and soft drinks) was banned while the sale of healthy food choices (e.g., fruits and vegetables, popcorn, 100% juice, nuts, dried fruits, sandwiches) were mandated. The implementation was less successful in private schools where the ministerial order was not mandatory and the school shop owners were primarily revenue-oriented.Parental involvement: parents’ attendance at meetings was impacted by various factors, namely when both parents worked, and when younger children were still at home. Parents’ presence was enhanced when they were given a direct motivation such as an invitation to a breakfast meal.

## 4. Discussion

The present study demonstrated that long-term multicomponent nutrition and physical activity intervention program can improve dietary behaviors, knowledge and self-efficacy scores of students aged 8–12 when administered over a two-year period; with a sustained effect following one-year washout; however, a lower effect on overweight and obesity was detected. In fact, findings from this study showed that the two-year intervention was effective in significantly reducing the odds of developing overweight or obesity after one-year washout only in public schools; students in intervention groups were 52% less likely to develop overweight or obesity compared to control after the three-year study period. In addition, no similar impact was observed when considering mean BMI z-scores.

These results corroborate the findings of research studies that observed a null effect on BMI after three years of intervention [21], whereas others showed a reduction in both the odds of developing obesity or overweight and in BMI z-scores, especially for obese or overweight students [22]. Recent reviews and meta-analyses have shown that interventions combining diet and physical activity might reduce adiposity in 6–12 year old children with low certainty evidence [10,23]. The non-significant reduction in BMI z-scores, in this study, could be attributed to the short follow-up period that may have affected the detection of effects on BMI [24]. The fact that our intervention included less physical activity sessions compared with other studies [10] may explain as well the lack of effect on mean BMI z-scores. Furthermore, Lebanese children live in an obesogenic environment (environments that promote high energy intakes and sedentary behavior) which might have affected BMI outcomes. In addition, the built environment in urban areas such as the city of Beirut and its suburbs prevents children from engaging in spontaneous physical activity. A body of evidence suggests that there is a link between the built environment, physical activity, obesity and chronic diseases [25]. Efforts to improve the urban environments and increase pedestrian friendly neighborhoods have been emerging in developed countries. Like other developing countries, improving the urban environment is not one of Lebanon priorities as the main concern of policy makers is around economic and political problems.

In the current study, the lower odds of developing overweight or obesity was mostly observed in students from public schools who are of relatively lower SES. It is important to note that in developing countries, in contrast with developed ones, there is a general trend of increasing the prevalence of obesity with increasing SES [26]. In Lebanon, for example, people belonging to high socio-economic classes have more access to unhealthy food intake and are more sedentary though they practice more leisure activities [27]. In addition, children from high SES have access to personal television, tablets, or computers [27]. This may explain the absence of a decrease in the odds of developing overweight or obesity among students of higher SES in private schools.

This intervention program had positive effects on students’ dietary behavior predominantly in public schools where a significant change in the consumption of fruits, vegetables and energy dense snacks was shown. The adoption and integration of the program’s components by MEHE within the public schools’ health curriculum could explain the positive effect of the intervention on the students’ socio-cognitive determinants (knowledge and self-efficacy scores) observed in public schools, while it was less apparent in private schools where no such integration was implemented. The process evaluation showed that the intervention was delivered consistently between all of the schools involved in the treatment group; however, in the second year, the fidelity of the intervention decreased in private schools, thus reducing both the reach and dose received by the students. Other studies have also reported that integrative approaches helped in reaching positive effects on students’ dietary behaviors [28].

The increased consumption of fruits and vegetables in public schools may have contributed to the observed positive effects on the odds of overweight or obesity. Fruits and vegetables consumption have been advocated for their positive impact on health and prevention of diet related non-communicable diseases including obesity. Literature reports recommend increased consumption of fruits and vegetables as beneficial components in weight management as it was associated with lower BMI [29,30].

Despite the increase in the odds of the fruits and raw vegetables consumption among students in private schools receiving the intervention, there was no reduction in chips consumption. It can be hypothesized that the absence of effect on overweight and obesity may be due to the fact that the increased intake of healthy food seen in students from private schools did not offset the harmful effect of continued consumption of unhealthy food on BMI. In our study, this was not verified since we did not calculate students’ total caloric intake. Reviews have shown that interventions resulting in changing to one healthy behavior do not necessarily lead to an overall favorable result on child’s health [23]. In addition, the adoption of Ajyal Salima program components by the MEHE may have facilitated the alteration of the school food environment in public schools. In fact, the MEHE has issued a ministerial order regulating the type of foods sold in public school shops. This may have resulted in the change in the consumption of energy-dense snacks and drinks in public schools, but not in private ones, where a similar measure was not implemented. An increase in the consumption of energy-dense snacks and drinks has been suggested as associated with increased energy intake and thus possibly responsible for the rise in obesity rates [31]. In fact, the availability of such snacks and drinks in schools could be related to students’ high intake of soft drinks, fat, and a lower intake of fruits and vegetables. The majority of studies with strategies aiming at changing the obesogenic environment such as altering the availability of snack foods in canteens, showed clear changes in either students diet or the quality of foods and beverages purchase [32,33].

Our intervention program focused as well on encouraging PA and decreasing sedentary behaviors among students; changes were detected after one-year washout in the total sample. The reduced number of targeted physical education sessions and the absence of sports specialists in our intervention may have hindered the improvement in students’ PA levels. Studies have shown that interventions focusing mainly on physical education had positive effects on PA levels in children but not necessarily on obesity rates. These studies concluded that school-based PA in child-friendly environments had positive impact on children’s daily PA during school days and weekends [34,35,36]. Thus, policies aiming at increasing physical education sessions at school across the curriculum or as extracurricular activities are good strategies towards increasing students’ daily physical activity.

This study had several limitations. The convenient selection of schools may have introduced selection bias; MEHE selected public schools while private schools were enrolled based on their willingness. The study was not initially powered to examine the moderating effect of schools, the impact of the intervention was explored separately for private and public, in addition to the total sample, to inform future research. Post-hoc adjustments for multiple comparisons were not conducted as we believe that the process oversimplifies a complex issue in hypothesis testing as discussed in the literature [37]. Nevertheless, even when the critical alpha was set at a more conservative value (*p* = 0.01), all the results remained the same except for sweet drinks (*p* = 0.032) and physical activity (*p* = 0.044). We have, therefore, provided in our manuscript the effect sizes, confidence intervals and *p*-values so that readers can make their own judgement about the relative weight of the conclusions.

Physical activity and dietary behaviors were measured using self-reported questionnaires, which remain subjective, give rise to social desirability and are prone to reporting error. More objective measures must be considered, mainly using tools to assess the intensity of PA activity in students. Another limitation is the absence of the total energy intake of students that would have contributed to more comprehensive assessment of caloric intake and informed any change in obesity rates. Parents’ participation in school meetings and activities was relatively low, and future interventions should focus on finding better ways to enhance parental involvement. At the end of the third year, 30% of the participants were lost to follow-up which may explain the high odds ratios obtained for certain values without the statistical significance. Finally, although our study was of three-year duration it was not long enough to detect changes in mean BMI z-scores.

## 5. Conclusions

The two-year school-based intervention to promote healthy eating and physical activity showed a positive and sustained effect on the adoption of healthy eating habits. Students in intervention schools reported significant increases in fruits and vegetables consumption post intervention with a sustained effect after one-year washout. However, the reduced odds of developing obesity was observed only after the one-year washout period in students from public schools but not from private schools. Moreover, our study did not detect changes in mean BMI z-scores. Comprehensive approaches, including the components used in this study, may instill skills for a long-term impact on children’s health, but there were also many lessons learnt that could inform future interventions.

Based on our study findings and general observations, we recommend that additional strategies such as national-level policy measures within schools are mandated and implemented for a greater success in obesity prevention programs, especially in developing countries. While school food environments and contexts differ based on cultural and structural differences [32], the present study can provide insightful evidence for similar policy changes in other countries of the Middle Eastern region to address the growing problem of childhood obesity in this part of the world.

Future studies should collect recent food consumption data on the participants and address additional strategies that can further impact outcome measures such as intervention time and follow-up period.

## Figures and Tables

**Figure 1 nutrients-12-02687-f001:**
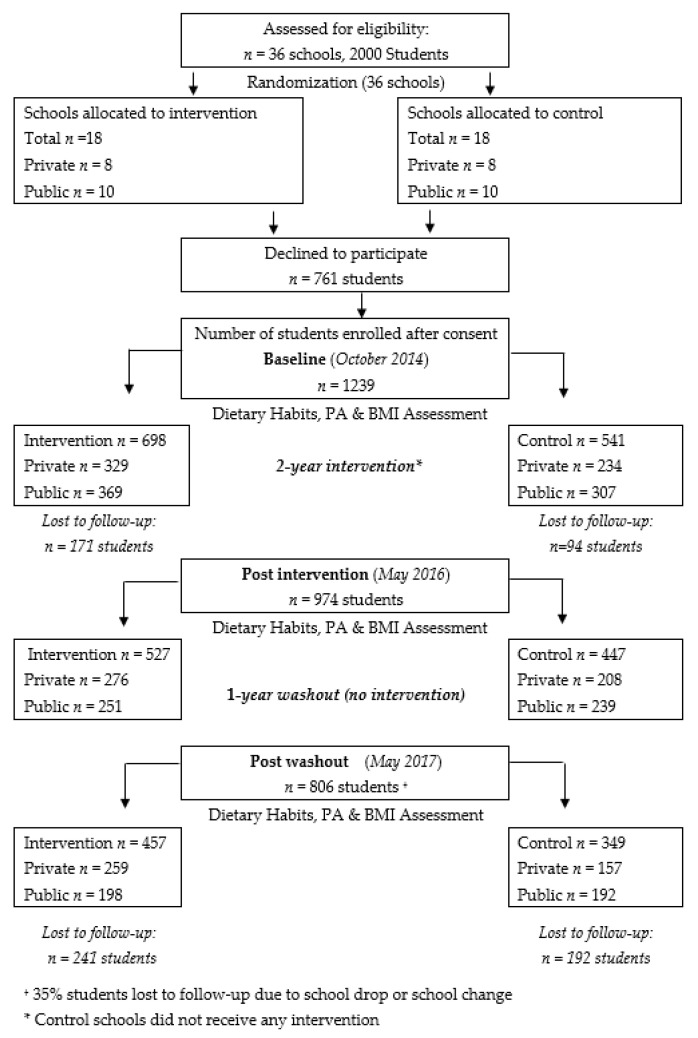
Flowchart illustrating students’ recruitment, sample retention and data collection in both groups from baseline to study completion.

**Table 1 nutrients-12-02687-t001:** Baseline characteristics of participating students in intervention and control groups.

	Total Sample	Intervention	Control	*p*-Value *
**No. of students**	1239	698	541	
**Mean Age (yrs.)**	9.95 ± 1.13	9.81 ± 0.68	10.13 ± 0.68	<0.01
	***n* (%)**			
**Gender:**				0.165
Male	574 (46.3)	311 (54.2)	263 (45.8)	
Female	666 (53.7)	387 (58.1)	279 (41.9)	
**Type of school:**				0.185
Private schools ^†^	564 (45.5)	329 (47.1)	235 (43.4)	
Public schools ^†^	676 (54.5)	369 (52.9)	307 (56.6)	
**Body mass index (BMI):**				0.235
Normal	737 (59.5)	400 (57.3)	337 (62.3)	
At risk	281 (22.7)	161 (23.1)	120 (22.2)	
Overweight	181 (14.6)	112 (16)	69 (12.8)	
Obese	40 (3.2)	25 (3.6)	15 (2.8)	
**Dietary and physical activity (PA) habits:**				
Fruits consumption ^§^	883 (71.9)	485 (54.9)	398 (45.1)	0.129
Raw vegetables consumption ^§^	809 (65.6)	467 (57.7)	342 (42.3)	0.137
Having chips ^§^	695 (56.4)	425 (61.2)	270 (38.8)	<0.001
Drinking soft drinks ^§^	744 (60.4)	425 (57.1)	319 (42.9)	0.404
Drinking sweet drinks ^§^	559 (45.4)	317 (56.7)	242 (43.3)	0.808
Exercise after school ^§§^	483 (39.2)	274 (56.7)	209 (43.3)	0.697

* *p*-values were derived from independent *t* test for continuous variables and chi square for categorical variables, ^†^ Private schools represent the medium to high socioeconomic status (SES); public schools represent the low SES, significant at *p* < 0.05; ^§^ Reference category: at least once per day versus less than once per day; ^§§^ At least 3 times per week versus less than 3 times per week.

**Table 2 nutrients-12-02687-t002:** Change in mean BMI Z-score in intervention and control groups relative to baseline at post-intervention and after washout.

	Post Intervention *N* = 974 Mean ± SE			After 1 Year Washout *N* = 806 Mean ± SE		
	Intervention *N* = 527	Control *N* = 447	*p*-Value *	Intervention *N* = 457	Control *N* = 349	*p*-Value *
**Total sample**	0.07 ± 0.047	0.145 ± 0.048	0.272	0.134 ± 0.048	0.237 ± 0.054	0.1623
**Private schools** ^†^	0.112 ± 0.072	0.136 ± 0.073	0.8136	0.111 ± 0.072	0.141 ± 0.083	0.795
**Public schools** ^†^	0.024 ± 0.063	0.152 ± 0.065	0.1713	0.164 ± 0.065	0.316 ± 0.066	0.1165

Controlling for age, gender and baseline BMI * Significant at *p* < 0.05 Values in this table represent means ± SE, after adjustment for age and baseline BMIZ, ^†^ Private schools represent the medium to high SES; public schools represent the low SES.

**Table 3 nutrients-12-02687-t003:** Odds ratio of overweight/obesity change in intervention vs. control groups¶ post-intervention and after washout.

	OR (95% CI) Post Intervention *n* = 974	*p*-Value *	OR (95% CI) after 1-Year Washout *n* = 806	*p*-Value *
**Total**	1.35 (0.77; 2.36)	0.284	0.79 (0.47; 1.32)	0.373
**Private schools** ^†^	1.39 (0.71; 2.70)	0.333	1.17 (0.75; 1.83)	0.49
**Public schools** ^†^	1.28 (0.53; 3.10)	0.58	0.48 (0.26; 0.88)	0.018

Controlling for age, gender and baseline BMI * Significant at *p* < 0.05 ^†^ Private schools represent the medium to high SES; public schools represent the low SES.

**Table 4 nutrients-12-02687-t004:** Odds ratio of dietary habits change in response to the nutrition intervention among students.

	OR (95% CI) Post Intervention *n* = 974	*p*-Value *	OR (95% CI) after 1-Year Washout *n* = 806	*p*-Value *
**Fruits consumption ^§^**				
Total sample	1.56 (1.18; 2.08)	0.002	1.28 (0.99; 1.65)	0.054
Private schools **^†^**	1.84 (1.30; 2.60)	0.001	1.04 (0.76; 1.43)	0.770
Public schools **^†^**	1.44 (0.89; 2.32)	0.135	1.66 (1.14; 2.42)	0.008
**Raw Veggies consumption ^§^**				
Total sample	1.79 (1.32; 2.43)	0.000	1.66 (1.25; 2.19)	0.000
Private schools **^†^**	2.32 (1.42; 3.80)	0.001	1.33 (0.90; 1.95)	0.141
Public schools **^†^**	1.47 (0.95; 2.28)	0.082	2.19 (1.59; 3.01)	0.000
**Having chips ^§^**				
Total sample	0.52 (0.32; 0.84)	0.008	0.86 (0.58; 1.26)	0.493
Private schools **^†^**	0.59 (0.32; 1.11)	0.103	1.08 (0.62; 1.92)	0.776
Public schools **^†^**	0.44 (0.24; 0.8)	0.007	0.70 (0.49; 1.02)	0.060
**Drinking soft drinks ^§^**				
Total sample	0.64 (0.42; 0.95)	0.027	0.68 (0.48; 0.97)	0.031
Private schools **^†^**	0.66 (0.36; 1.23)	0.190	0.69 (0.39; 1.23)	0.213
Public schools **^†^**	0.59 (0.34; 1.02)	0.057	0.68 (0.44; 1.05)	0.081
**Drinking sweet drinks ^§^**				
Total sample	0.62 (0.40; 0.96)	0.032	0.71 (0.47; 1.07)	0.104
Private schools **^†^**	0.58 (0.28; 1.19)	0.136	0.90 (0.48; 1.69)	0.752
Public schools **^†^**	0.63 (0.37; 1.07)	0.087	0.62 (0.39; 1.00)	0.048
**Exercise after school ^§§^**				
Total sample	1.25 (0.87; 1.80)	0.225	1.39 (1.00; 1.92)	0.044
Private schools **^†^**	1.07 (0.64; 1.78)	0.782	1.47 (0.94; 2.30)	0.087
Public schools **^†^**	1.44 (0.87; 2.41)	0.154	1.27 (0.77; 2.09)	0.330

Odds ratio (OR) controlled for baseline measures, age and gender * Significant at *p* < 0.05. ^†^ Private schools represent the Medium to high SES; Public schools represent the low SES. ^§^ Reference category: at least once per day versus less than once per day; ^§§^ At least 3 times per week versus less than 3 times per week.

**Table 5 nutrients-12-02687-t005:** Change in average knowledge and self-efficacy scores in intervention vs. control at post-intervention and after washout.

	Post Intervention		*p* Value *	After 1-Year Washout		*p* Value *
	Intervention	Control		Intervention	Control	
	*n* = 528	*n* = 448		*n* = 457	*n* = 349	
Mean Difference (95% CI)				Mean Difference (95% CI)		
**Knowledge score**						
Total sample	3.13 (2.80; 3.46)	1.37 (1.02; 1.71)	<0.001	2.84 (2.47; 3.20)	1.57 (1.16; 1.98)	<0.001
Private schools ^†^	2.87 (2.29; 3.45)	1.16 (0.57; 1.75)	<0.001	2.44 (1.83; 3.04)	1.31 (0.56; 2.05)	0.014
Public schools ^†^	3.41 (3.01; 3.81)	1.54 (1.13; 1.95)	<0.001	3.35 (2.91; 3.79)	1.78 (1.33; 2.23)	<0.001
**Self-efficacy score**						
Total sample	1.16 (0.76; 1.56)	0.29 (−0.13; 0.70)	0.003	0.68 (0.17; 1.20)	−0.12 (0.69; 0.43)	0.031
Private schools ^†^	1.15 (0.68; 1.62)	0.51 (−0.01; 1.03)	0.050	0.44 (−0.13; 1.01)	0.33 (−0.39; 1.05)	0.779
Public schools ^†^	1.17 (0.45; 1.89)	0.09 (−0.65; 0.83)	0.030	1.00 (0.12; 1.88)	−0.50 (−1.43; 0.42)	0.016

* Significant at *p* < 0.05 + controlling for baseline measures and age and gender ^†^ Private schools represent the medium to high SES; public schools represent the low SES.

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
