# Peer review of "Impact of a Three-Year Obesity Prevention Study on Healthy Behaviors and BMI among Lebanese Schoolchildren: Findings from Ajyal Salima Program"

_nutrients, 2020, doi:10.3390/nu12092687_

Round 1

Reviewer 1 Report

Thank you for the opportunity to review this paper. Strengths include focus on an important topic (childhood obesity) and understudied population (Lebanese youth), relatively large sample (n=806), outcome assessment both immediately post-intervention and at 1 year washout, exploration of effects by SES subgroup, and consideration of both behavioral and anthropometric outcomes. Opportunities for improvement are described below.

Introduction

*Paragraph 2: Can the authors provide the prevalence of childhood obesity in Lebanon? They state that rates are increasing, but no specific prevalence is provided.

*Line 53: The authors briefly mention factors associated with childhood obesity intervention effectiveness, citing the associated Cochrane review. However there is an extensive literature specific to school-based obesity interventions (including multiple review papers), which the authors may want to include.

*Paragraph 3: I am a bit unclear on the relationship between the HEALTH-E-PALS results described in paragraph 3 and the current paper. I assume that the HEALTH-E-PALS was piloted in other schools, evaluated and published (reported in citation #9), and then scaled up and increased in duration to two years (reported in the current paper). However a program called Healthy Kids is mentioned on line 161. Can the authors provide more clarity?

Materials and Methods

*Data collection (line 119): Can the authors provide more detail about anthropometric measurement (e.g., shoes off, wall-mounted stadiometer, scale type, citation for methods if available)?

*Data collection (line 142): Can the authors provide examples of the self-efficacy questions, as they do for the dietary habits and physical activity questions? Also, was general self-efficacy assessed or self-efficacy for obesity-related behaviors (e.g., self-efficacy for eating healthy, self-efficacy for physical activity)?

*Process evaluation: Can the authors provide more detail (e.g., how often were fidelity checks completed, was every school assessed, how were data synthesized, etc.)?

*Intervention design (section 2.5): Can the authors provide more detail about the specific intervention activities? For example, the second component (involving parents) is known to be critically important for childhood obesity interventions, yet is only described in one sentence.

*Intervention design (line 175): “In 2014, the Ajyal Salima programme was adopted by the MEHE and incorporated into the 175 curriculum of the Health Education Unit.” It is unclear what is meant by this sentence.

*Data analysis: It is unclear whether clustering was accounted for in the regression models. The authors state that clustering was accounted for, but also state that they used general linear models (rather than multi-level models or GEEs). Can the authors please clarify?

*Data analysis (line 193): Not only dietary behaviors, but other outcomes (e.g., physical activity, self-efficacy), were examined too, correct?

Results:

*Process evaluation: This section of the results seems underdeveloped, and it is unclear, based on the methods, how the data was collected and synthesized. For example, barriers to and facilitators of parent attendance are described, but it is unclear whether that was assessed via informal conversation, survey, qualitative interviews, asking teachers, etc.

Discussion

*Paragraph 1: The synthesis seems a bit overenthusiastic, given that BMIz change was not different and obesity/overweight was only different at washout and only for public schools. The authors may want to temper. Small/lack of effect on BMI is consistent with many school-based interventions.

*Lines 286-289: The authors describe that the short follow up may be a reason for lack of effect, but any effect would likely be seen within 2 years (including 1 year post-intervention). The authors also mention lack of a physical activity component. What about other reasons? Can the authors provide more depth, considering the extensive literature on school-based interventions? One suggestion - perhaps the changes in food environment and increase in sedentary lifestyles (described in introduction paragraph 2) are too impactful and “cancel out” any effect of a low-intensity program based primarily in the school.

Overall

*Check formatting throughout - there are a few paragraphs that lack indentation.

Reviewer 2 Report

This study investigated the long-term effects of a school-based intervention program when implemented over two years on Body Mass Index, healthy dietary behaviors, and physical activity; and if the effects are sustained after one-year washout.

This is valuable research in the light of childhood overweight and obesity policies long-term effectiveness. This article scientifically sounds, nevertheless, I have some concerns that I think they will improve your paper.

Abstract

It is complete and well structured. It provides the article’s main information about background, methods, results and conclusions.

Introduction

The introduction is exhaustive and provides a complete overview of the matter.

The first paragraph mentions the questions and issues that outline the background of the study and establishes, the context and relevance of the problem.

The second paragraph includes the importance of the problem and unclear issues.

The last paragraph states the main objective and the questions addressed in the manuscript.

Suggestions:

L36-39 Data are from 2013, they should be updated.

L 47-51 You should focus on children, not on adolescents.

Materials and Methods

Methods are adequately described. This information provides to the journal audience a better and complete understanding of the manuscript.

It should be noted the useful scheme (Figure 1) to better understand all the work system.

Suggestion:

L139-140  “correct/incorrect” answer should be changed to “positive/negative” answer.

Results

This section includes the necessary data of the experimental results. All tables and figures are clear and have a short explanatory title and caption.

Conclusions

Although this is not a mandatory section, the authors have included it. It provides a clear idea of the main results of the investigation.

References

Please check references: journal names must be in italics.

I am looking forward to reading the final version of the manuscript.

Round 2

Reviewer 1 Report

Thank you for addressing the reviewer comments.